# A DNA Replication Mechanism Can Explain Structural Variation at the Pigeon *Recessive Red* Locus

**DOI:** 10.3390/biom12101509

**Published:** 2022-10-18

**Authors:** Jonathan Haddock, Eric T. Domyan

**Affiliations:** Department of Biology, Utah Valley University, Orem, UT 84058, USA

**Keywords:** structural variation, DNA replication, FoSTeS/MMBIR

## Abstract

For species to adapt to their environment, evolution must act upon genetic variation that is present in the population. Elucidating the molecular mechanisms that give rise to this variation is thus of crucial importance for understanding how organisms evolve. In addition to variation caused by point mutations, structural variation (deletions, duplications, inversions, translocations) is also an important source of variety. Mechanisms involving recombination, transposition and retrotransposition, and replication have been proposed for generating structural variation, and each are capable of explaining certain rearrangements. In this study, we conduct a detailed analysis of two partially overlapping rearrangements (*e^1^* and *e^2^* allele) in domestic rock pigeon (*Columba livia)* which are both associated with the *recessive red* phenotype. We find that a replicative mechanism is best able to explain the complex architecture of the *e^1^* allele, and is also compatible with the simpler architecture of the *e^2^* allele as well.

## 1. Introduction

Relatively little is known about the underlying mechanisms responsible for generating structural changes responsible for a great deal of both evolutionary- and medically- relevant variation. One prominent model is non-allelic homologous recombination (NAHR), which is when crossing over occurs between different genomic regions [1]. NAHR can be mediated by either low-copy repeats (LCRs) or repetitive elements (such as transposons) in the genome, which may cause chromosomes to mis-pair during meiotic synapse. A second model, non-homologous end joining (NHEJ), occurs when double-strand breaks occur in DNA and repair is initiated between the two broken ends [2]. Mobile DNA elements such as transposons and retrotransposons also generate structural variation by inserting DNA in novel locations or orientations [3,4].

A DNA replication-based mechanism, known as fork-stalling and template switching/microhomology-mediated break-induced replication (FoSTeS/MMBIR) is distinct from the others in that is capable of explaining certain complex rearrangements that non-replication based mechanisms cannot [5,6]. In this mechanism, an active replication fork can stall due to some DNA lesion, and switch templates using short (e.g., several-nucleotide) stretches of microhomology to re-initiate DNA replication. Depending on whether the new template strand is behind, ahead, or on the reverse complement of the original template strand, duplications, deletions, and inversions can all occur. Similar to NAHR, FoSTeS/MMBIR events can also be mediated by LCRs, as these repeats can generate secondary structures that predispose the replication machinery to errors. However, because multiple FosTeS/MMBIR events may occur during a single replication cycle, this mechanism can provide a more parsimonious explanation of complex rearrangements than the previously mentioned models can.

Understandably so, the vast majority of research into naturally occurring structural variation are human studies [5,6,7,8] although several involve non-human models, including recurrent *Pitx1* enhancer deletions in sticklebacks [9,10], retrotransposon-mediated gene duplication in tomato [3], and transposition associated with a color polymorphism in cichlids [4]. Identifying additional instances of structural variation in free-living organisms, particularly those that alter the organisms’ phenotype, and determining what molecular mechanisms may have caused them will ultimately provide a greater understanding of genomic mechanisms of evolution in general.

Previously, we identified two partially overlapping deletions/rearrangements upstream of *Sox10* as the molecular identity of the classical *recessive red* locus in pigeon (*e^1^* and *e^2^* alleles) [11]. Both alleles contain a deletion of a melanocyte-specific enhancer that causes downregulation of *Sox10*, ultimately resulting in the production of red pheomelanin instead of black eumelanin [12]. The *e^1^* allele consists of a 7 kb rearrangement containing several deletions interspersed with inversions and duplications, while the *e^2^* allele consists of a simple 2.5 kb deletion. To better understand the mechanism that may have led to these rearrangements we performed detailed sequence analysis and find that FoSTeS/MMBIR can explain the complex structural variation of the *e^1^* allele and is also compatible with the *e^2^* allele.

## 2. Materials and Methods

The *Columba livia* reference genome was generated from a bird homozygous for the *e^1^* allele [11,13]. The rearrangement is located on scaffold 974 from 568,826 to 569,523 in Cliv_1.0, and on AKCR02000009.1 from 3,610,268 to 3,610,965 in Cliv_2.1 [14], though the complementary strands of this region are annotated between the two assemblies. The wild-type *E^+^* and mutant *e^2^* alleles are published as GenBank Accessions KJ023252 and KJ023253, respectively. Because the strands published in KJ023252 and KJ023253 are the same as the annotated strand in Cliv_1.0, that assembly is used as reference for these data.

Approximately 700 bp proximal to the rearrangement of the *e^1^* allele, Cliv_1.0 has a gap. Using DNA samples from two birds homozygous for each allele as template, we PCR-amplified across the gap using primers (5′-GCATGGTGACCTTTGACCTT-3′ and 5′-GGGGAAAAATGCTCGTGTAA-3′) and Sanger-sequenced the PCR products. After filling in the gap with the results of the Sanger read, we used RepeatMasker v. 4.1.2 with default parameters [15] to identify repetitive elements in each allele and BLASTZ v. 1.0 [16] to individually align each of the mutant alleles to the wild-type allele.

## 3. Results

Because of the close proximity of a gap in the reference genome to the location of the *recessive red* mutations, we Sanger-sequenced across this gap, and found that the sequence was identical among all three alleles (Appendix A). RepeatMasker identified some sequence showing similarity to a SINE/MIR element, but this sequence was identical across all three alleles and did not occur at any of the rearrangement breakpoints (Appendix A). No other putative transposon or retrotransposon sequence was identified in the analyzed regions.

After aligning the alleles using BLASTZ, we found that the *e^1^* allele is a complex rearrangement consisting of, in sequence; a 3.6 kb deletion, a ~400 bp inversion (Inversion 1), a 2 kb deletion, two inversions of ~200 and 70 bp (Inversions 2 and 3, respectively), and an ~800 bp deletion which spans the mcs7 melanocyte enhancer [17] (Figure 1a, Appendix A). The three inverted segments have maintained their original proximo-distal order relative to their original location in the E*^+^* allele. Inversions 2 and 3 also share a 56 bp duplication. The only sequence unique to *e^1^* relative to *E^+^* are insertions of “TC” at either end of the rearrangement.

While the beginning and end of the rearrangement in the *e^1^* allele contain dinucleotide insertions, the junctions of the inversions within the rearrangement contain microhomologies characteristic of FoSTeS/MMBIR events. A microhomology of “CA” is present at the junction of Inversion 1 and Inversion 2, and a microhomology of “CTGA” is present at the junction of Inversion 2 and Inversion 3 (Figure 1a, Appendix A).

The *e^2^* allele, in contrast, is a single ~2.8 kb deletion which also spans the mcs7 melanocyte enhancer (Figure 1b, Appendix A). The proximal end of the deletion is located within a 90-nucleotide repetitive region made up entirely of purines (A/G). This allele contains an “AG” microhomolgy at the junction of the deletion, also consistent with FoSTeS/MMBIR.

## 4. Discussion

An important avenue of research to elucidate how organisms evolve is to understand the molecular mechanisms that generate genetic diversity; including both point mutations and larger structural rearrangements. In this study we describe the molecular structure of two alleles in domestic rock pigeon that cause a change in plumage color. We find that for both alleles, the FoSTeS/MMBIR model can offer a potential explanation for their formation. For the *e^1^* allele, the entire rearrangement could have occurred during a single round of replication that had 4 FoSTeS events (Figure 1c–f). The structure of the rearrangement suggests that a lesion may have caused the polymerase of the leading strand of one replication fork to disengage and invade the lagging strand of either the same or another replication fork (Figure 1c). As the replication fork proceeded distally, multiple rounds of synthesis, disengaging, and re-invasion of the lagging strand template could have generated the three inversions in the observed orientation. The duplication contained in Inversion 2 and 3 could have been caused if DNA polymerase had used the same section of template DNA to synthesize the 5′ end of Inversion 2 (Figure 1d) and the 3′ end of Inversion 3 (Figure 1e) as the replication fork proceeded distally. Finally, DNA polymerase may have disengaged the lagging strand and re-engaged the original leading strand template, 7.5 kb distal to where it first disengaged (Figure 1f). The mechanism causing the insertion of “TC” at either end of the rearrangement is unclear, but could be due to strand breakage and repair.

In contrast to the *e^1^* allele, the *e^2^* allele is a simple deletion, but the presence of the 5′ deletion breakpoint within a 90-nucleotide long stretch of purines and the “AG” microhomology at the deletion junction is also compatible with the FoSTeS/MMBIR model. To generate the *e^2^* allele, DNA polymerase may have slipped off of the template strand and re-engaged with the same strand approximately 2700 nucleotides later, using a two-nucleotide span of microhomology to reinitiate synthesis.

The unique breakpoints of each *recessive red* allele indicate that each rearrangement occurred independently; furthermore, a deletion that spans this same region is also found in chicken [18]. This raises the question as to whether this region of the avian genome is particularly prone to rearrangements or whether these rearrangements were simply more likely to be identified and artificially selected due to their effect on feather color. The presence of LCRs can create complex genome architecture that may make FoSTeS/MMBIR events more likely [7]. In addition, certain regions of the genome may contain DNA fragility that increase the likelihood of breakage and repair [10]. An important area of future research will be to determine whether these or other features exist in this region of the avian genome that could predispose it to rearrangements.

## 5. Conclusions

Mutation provides the raw material for evolution by natural selection; therefore understanding mutational mechanisms is crucial for understanding how organisms adapt to their environment. Given that copy number variation likely contributes more to genetic variability on a per-nucleotide level than SNPs and small indels combined [19], understanding the mechanisms leading to copy number variation is of crucial importance. For the *recessive red* locus in pigeons, we find that the FoSTeS/MMBIR model provides a plausible mechanism to explain the observed rearrangements and provides support for the idea that replicative events can contribute to phenotypic variation in diverse organisms. As additional cases of structural variation in free-living organisms are identified, it will be important to determine the possible mechanisms generating them as well.

## Figures and Tables

**Figure 1 biomolecules-12-01509-f001:**
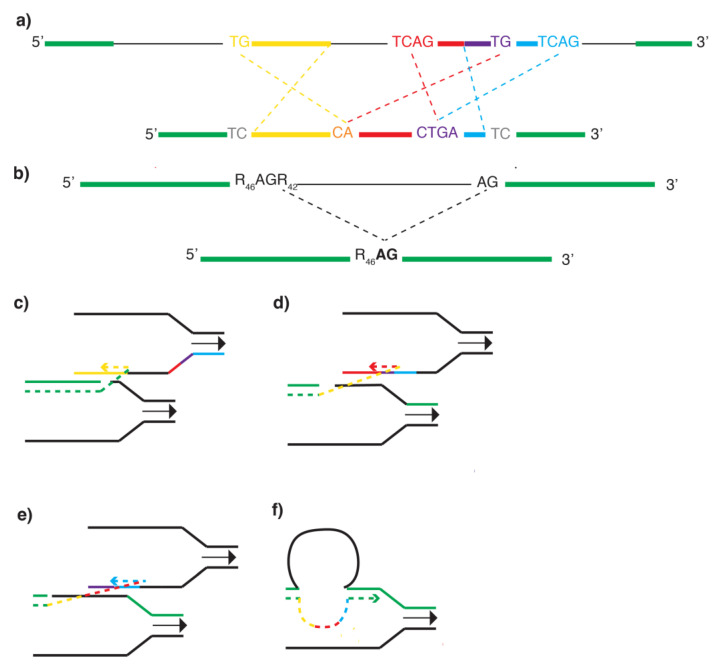
Structure of the *recessive red* alleles. (**a**) Illustration of the *E^+^* (top) and *e^1^* (bottom) alleles in relation to one another. DNA in the same orientation in green. Deleted DNA as thin black line. TC insertions in grey font. Inversion 1 in yellow, Inversion 2 in red, and Inversion 3 in blue. DNA duplicated in Inversions 2 and 3 in purple. Sequence of microhomologies at the inversion junctions shown. (**b**) Illustration of the *E^+^* (top) and *e^2^* (bottom) alleles in relation to one another. DNA in same orientation in both alleles in green. Deleted DNA as thin black line. Proximal deletion breakpoint is in a run of 90 purines (R), the junction of the deletion forms a microhomology as shown. (**c**–**f**) Putative replicative mechanism to explain formation of *e^1^ allele.* As replication fork (black) proceeds left-to-right, a lesion in the template strand (break in green line) may cause DNA polymerase to disengage from leading strand template and invade the lagging strand template. FoSTeS event 1 would cause Inversion 1 in yellow. As the replication fork proceeds distally, subsequent disengagement and re-engagement with the lagging strand template would produce Inversion 2 (red, **d**) and 3 (blue, **e**), before re-engagement with the original template (green, **f**) eventually occurs.

## Data Availability

The wild-type *E^+^* and mutant *e^2^* alleles are published as GenBank Accessions KJ023252 and KJ023253, respectively.

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
