# Peer review of "A DNA Replication Mechanism Can Explain Structural Variation at the Pigeon Recessive Red Locus"

_biomolecules, 2022, doi:10.3390/biom12101509_

Round 1

Reviewer 1 Report

This study investigates the variation underlying 2 different recessive alleles for color in pigeons. They show that structural variation at a SOX10 enhancer is responsible. Moreover, they document evidence for a Complex Genomic Rearrangement best explained by FoSTeS/MMBIR providing a very nice example of the role of this mutational mechanism in evolution and trait manifestation.                            Jim Lupski

Author Response

We thank Reviewer 1 for their comments. There were no suggested revisions from this reviewer.

Reviewer 2 Report

The author reports sequence analysis on two alleles associated with recessive red phenotype in domestic rock pigeon (Columba livia). From this they speculate that the complexities in one allele (e2) may have arisen from an error during DNA replication due to fork-stalling and template switching/microhomology-mediated break-induced replication (FoSTeS/MMBIR). The proposal is reasonable though it lacks any experimental verification. The paper suffers from weak sentence structures and misuse of the word “catalyzed.”

The lead sentence is a case in point with a trailing “it”.

“Structural variation is responsible for a great deal of both evolutionarily- and medically-relevant variation; yet relatively little is known about the underlying mechanisms responsible for generating it.”

A better sentence would be: 

“Relatively little is known about the underlying mechanisms responsible for generating structural changes responsible for a great deal of both evolutionarily- and medically-relevant variation.”

Misuse of “catalyzed”

“NAHR can be catalyzed by either low-copy repeats (LCRs) or repetitive elements (such as transposons)….

“Similar to NAHR, FoSTeS/MMBIR events can also be catalyzed by LCRs….”

LCRs are not catalytic entities.  Enzymes and ribozymes catalyze reaction. Gene sequences to not. Perhaps the better word is “mediate.”
